# Improving GAN Training with Probability Ratio Clipping and Sample Reweighting

Yue Wu[1], Pan Zhou[2], Andrew Gordon Wilson[3], Eric P. Xing[1,4], Zhiting Hu[1,5]
[1]Carnegie Mellon University, [2]National University of Singapore, [3]New York University
[4]Petuum Inc., [5]UC San Diego
{ywu5, epxing}@andrew.cmu.edu, pzhou@u.nus.edu, andrewgw@cims.nyu.edu, zhitinghu@gmail.com

## Abstract

Despite success on a wide range of problems related to vision, generative adversarial networks (GANs) often suffer from inferior performance due to unstable training, especially for text generation. To solve this issue, we propose a new variational GAN training framework which enjoys superior training stability. Our approach is inspired by a connection of GANs and reinforcement learning under a variational perspective. The connection leads to (1) probability ratio clipping that regularizes generator training to prevent excessively large updates, and (2) a sample re-weighting mechanism that improves discriminator training by downplaying bad-quality fake samples. Moreover, our variational GAN framework can provably overcome the training issue in many GANs that an optimal discriminator cannot provide any informative gradient to training generator. By plugging the training approach in diverse state-of-the-art GAN architectures, we obtain significantly improved performance over a range of tasks, including text generation, text style transfer, and image generation.[1]

## 1 Introduction

Generative adversarial networks (GANs) [13] have achieved remarkable success in image and video synthesis [4, 32, 39]. However, it is usually hard to train a GAN well, because the training process is commonly unstable, subject to disturbances and even collapses. To alleviate this issue, substantial efforts have been paid to improve the training stability from different perspectives, e.g., divergence minimization [36, 37], Wasserstein distance with Lipschitz continuity of the discriminator [2, 15, 52], energy-based models [3, 58], to name a few.

In spite of the above progresses, the instability in training has not been well resolved [8], since it is difficult to well balance the strength of the generator and the discriminator. What is worse, such an instability issue is exacerbated in text generation due to the sequential and discrete nature of text [6, 12, 22]. Specifically, the high sensitivity of text generation to noise and the underlying errors caused by sparse discriminator signals in the generated text can often result in destructive updates to both generator and discriminator, enlarging the instability in GANs.

In this work, we develop a novel variational GAN training framework to improve the training stability, which is broadly applicable to GANs of a variety of architectures for image and text generation. This training framework is derived from a variational perspective of GANs [24] and the resulting connections to reinforcement learning (in particular, RL-as-inference) [1, 44] and other rich literature [5, 14, 23]. Our approach consists of two stabilization techniques, namely, probability ratio clipping and sample re-weighting, for stabilizing the generator and discriminator respectively. **(1)** Under the variational perspective, the generator update is subject to a KL penalty on the change

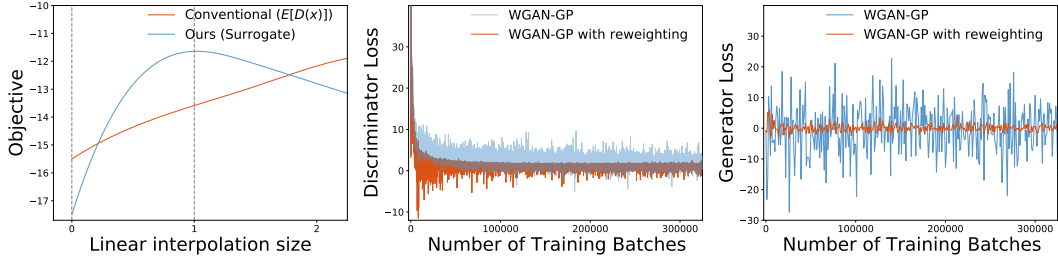

Figure 1: Illustration of the proposed approach for stabilizing GAN training. Results are from the CIFAR-10 experiment in Sec.4.1. **Left:** The conventional and surrogate objectives for generator training, as we interpolate between the initial generator parameters $\boldsymbol{\theta}_{old}$ and the updated generator parameters $\boldsymbol{\theta}_{new}$ which we compute after one iteration of training. The $\boldsymbol{\theta}_{new}$ obtains maximal surrogate objective. The surrogate objective curve starts decreasing after $x = 1$, showing the objective imposes a penalty for having too large of a generator update. In contrast, the conventional objective (for WGAN-GP) keeps increasing with larger generator updates. **Middle and right:** Discriminator and generator losses w/ and w/o sample re-weighting. WGAN-GP with our re-weighting plugged in shows lower variance in both discriminator and generator losses throughout training.

of the generator distribution. This KL penalty closely resembles that in the popular Trust-Region Policy Optimization (TRPO) [43] and its variant, i.e., Proximal Policy Optimization (PPO) [44]. This connection motivates a simple surrogate objective with a clipped probability ratio between the new generator and the old one. The probability ratio clipping discourages excessively large generator updates, and has shown to be effective in the context of stabilizing policy optimization [44]. Figure 1 (left) shows the intuition about the surrogate objective, where we can observe the objective value decreases with an overly large generator change and thus imposes regularization on the updates.

**(2)** When updating the discriminator, the new perspective induces an importance sampling mechanism, which effectively re-weights fake samples by their discriminator scores. Since low-quality samples tend to receive smaller weights, the discriminator trained on the re-weighted samples is more likely to maintain stable performance, and in turn provide informative gradients for subsequent generator updates. Figure 1 (middle/right) demonstrates the effect of the re-weighting in reducing the variance of both discriminator and generator losses.

Besides, our variational GAN training framework can provably overcome the training issue [59] that an optimal discriminator cannot provide any informative gradient to training generator. This issue usually occurs in GAN training [59], since the discriminator often converges much faster than the generator. Empirically, we conduct extensive experiments on a wide range of tasks, including text generation, text style transfer, and image generation. Our approach shows significant improvement over state-of-the-art methods, demonstrating its broad applicability and efficacy.

## 2 Related Work

**Wasserstein distance, WGAN, and Lipschitz continuity.** The GAN framework [13] features two components: a generator $G_\theta$ that synthesizes samples $\boldsymbol{x}$ given some noise source $\boldsymbol{z}$, namely $\boldsymbol{x} = G_\theta(\boldsymbol{z})$ with $\boldsymbol{z} \sim p_z(\boldsymbol{z})$, and a discriminator that distinguishes generator's output and real data, which provides gradient feedback to improve the generator's performance. WGAN [2] improves the training stability of GANs by minimizing the Wasserstein distance $W(p_r, p_\theta)$ between the generation distribution $p_\theta$ (induced from $G_\theta$) and the real data distribution $p_r$. Its training loss is formulated as:

$$\min_{\boldsymbol{\theta}} \max_{f \in \mathcal{D}} \mathbb{E}_{\boldsymbol{x} \sim p_r}[f(\boldsymbol{x})] - \mathbb{E}_{\boldsymbol{x} \sim p_\theta}[f(\boldsymbol{x})], \tag{1}$$

where $\mathcal{D}$ is the set of 1-Lipschitz functions; $f$ acts as the discriminator and is usually implemented by a neural network $f_\phi$. The original resort to enforce the Lipschitz constraint is through weight clipping [2]. WGAN-GP [15] later improves it by replacing it with a gradient penalty on the discriminator. CT-GAN [52] further imposes the Lipschitz continuity constraint on the manifold of the real data $\boldsymbol{x} \sim p_r$. Our approach is orthogonal to these prior works and can serve as a drop-in replacement to stabilize generator and discriminator in various kinds of GANs, such as WGAN-GP and CT-GAN.

Research on the Lipschitz continuity of GAN discriminators have resulted in the theory of "informative gradients" [59, 60]. Under certain mild conditions, a Lipschitz discriminator can provide

informative gradient to the generator in a GAN framework: when $p_\theta$ and $p_r$ are disjoint, the gradient $\nabla f^*(\boldsymbol{x})$ of optimal discriminator $f^*$ w.r.t each sample $\boldsymbol{x} \sim p_\theta$ points to a sample $\boldsymbol{x}^* \sim p_r$, which guarantees that the generation distribution $p_\theta$ is moving towards $p_r$. We extend the informative gradient theory to our new case and show theoretical guarantees of our approach.

**Reinforcement learning as inference.** Casting RL as probabilistic inference has a long history of research [1, 9, 10, 29, 40]. For example, Abdolmaleki et al. [1] introduced maximum a-posteriori policy optimization from a variational perspective. Tan et al. [48] connected the formulation with other paradigms of learning such as maximum likelihood estimation and data augmentation [20]. TRPO [43] is closely related to this line by using a KL divergence regularizer to stabilize standard RL objectives. PPO [44] further proposed a practical clipped surrogate objective that emulates the regularization. Our approach draws on the connections to the research, particularly the variational perspective and PPO, to improve GAN training.

**Other related work.** Importance re-weighting has been adopted in different problems, such as learning knowledge constraints [24], and improving VAEs [5] and GANs [23, 46]. We derive from the variational perspective which leads to re-weighting and clipping in the new context of GAN training stabilization. Our approach is orthogonal to and can be combined with other stabilization techniques such as large-batch training [4] and parameter averaging (EMA) [4, 55].

# 3 Improving GAN Training

## 3.1 Motivations

Our approach is motivated by connecting GAN training with the well-established RL-as-inference methods [1, 29, 48] under a variational perspective. The connections enable us to augment GAN training with existing powerful probabilistic inference tools as well as draw inspirations from the rich RL literature for stable training. In particular, the connection to the popular TRPO [43] and PPO [44] yields the probability ratio clipping in generator training that avoids destructive updates (Sec.3.2), and the application of importance sampling estimation gives rise to sample re-weighting for adaptive discriminator updates (Sec.3.3). The full training procedure is summarized in Alg.1.

Specifically, as described in Sec.2, the conventional WGAN formulation for updating the generator $p_\theta(\boldsymbol{x})$ maximizes the expected discriminator score $\mathbb{E}_{p_\theta}[f_\phi(\boldsymbol{x})]$, where $f_\phi$ is the Lipschitz-continuous discriminator parameterized with $\phi$. The objective straightforwardly relates to policy optimization in RL by seeing $p_\theta$ as a policy and $f_\phi$ as a reward function. Thus, inspired by the probabilistic inference formulations of policy optimization [1, 24, 48], here we transform the conventional objective by introducing a non-parameterized auxiliary distribution $q(\boldsymbol{x})$ and defining a new variational objective:

$$\mathcal{L}(\boldsymbol{\theta}, q) = \mathbb{E}_q[f_\phi(\boldsymbol{x})] - \mathrm{KL}\left(q(\boldsymbol{x}) \| p_\theta(\boldsymbol{x})\right), \tag{2}$$

where KL is the KL divergence. Intuitively, we are maximizing the expected discriminator score of the auxiliary $q$ (instead of generator $p_\theta$), and meanwhile encouraging the generator to stay close to $q$. We note that Hu et al. [24] have also related the above objective to GANs, with the different goal of integrating structured knowledge with deep generative modeling.

As we shall see in more details shortly, the new formulation allows us to take advantage of off-the-shelf inference methods, which naturally leads to new components to improve the GAN training. Maximization of the above objective is solved by the expectation maximization (EM) algorithm [34] which alternatingly optimizes $q$ at E-step and optimizes $\boldsymbol{\theta}$ at M-step. More specifically, at each iteration $t$, given the current status of generator parameters $\boldsymbol{\theta} = \boldsymbol{\theta}^{(t)}$, the E-step that maximizes $\mathcal{L}(\boldsymbol{\theta}^{(t)}, q)$ w.r.t $q$ has a closed-form solution:

$$q^{(t)}(\boldsymbol{x}) = \frac{p_{\theta^{(t)}}(\boldsymbol{x}) \exp\{f_\phi(\boldsymbol{x})\}}{Z_\phi}, \tag{3}$$

where $Z_\phi = \int_x p_{\theta^{(t)}}(\boldsymbol{x}) \exp\{f_\phi(\boldsymbol{x})\}$ is a normalization term that depends on the discriminator parameters $\phi$. We elaborate on the M-step in the following subsections, where we continue to develop the practical procedures for updating the generator and the discriminator, respectively.

## 3.2 Generator Training with Probability Ratio Clipping

The M-step optimizes $\mathcal{L}(\boldsymbol{\theta}, q^{(t)})$ w.r.t $\boldsymbol{\theta}$, which is equivalent to minimizing the KL divergence term in Eq.(2). However, since the generator $p_\theta$ in GANs is often an *implicit* distribution that does not permit evaluating likelihood, the above KL term (which involves evaluating the likelihood of samples from $q$) is not applicable. We adopt an approximation, which has also been used in the classical wake-sleep algorithm [18] and recent work [24], by minimizing the *reverse* KL divergence as below. With Eq.(3) plugged in, we have:

$$\min_\theta \text{KL}\left(p_\theta(\boldsymbol{x}) \| q^{(t)}(\boldsymbol{x})\right) = \min_\theta -\mathbb{E}_{p_\theta}\left[f_\phi(\boldsymbol{x})\right] + \text{KL}\left(p_\theta(\boldsymbol{x}) \| p_{\theta^{(t)}}(\boldsymbol{x})\right). \tag{4}$$

As proven in the appendix, this reverse KL approximation does not change the optimization problem in Eq.(2). The first term on the right-hand side of Eq.(4) recovers the conventional objective of updating the generator. Of particular interest is the second term, which is a new KL regularizer between the generator $p_\theta$ and its "old" state $p_{\theta^{(t)}}$ from the previous iteration. The regularizer discourages the generator from changing too much between updates, which is useful to stabilize the stochastic optimization procedure. The regularization closely resembles to that of TRPO/PPO, where a similar KL regularizer is imposed to prevent uncontrolled policy updates and make policy gradient robust to noises. Sec.3.4 gives analysis on the KL-regularized generator updates.

In practice, directly optimizing with the KL regularizer can be infeasible due to the same difficulty with the implicit distribution as above. Fortunately, PPO [44] has presented a simplified solution that emulates the regularized updates using a clipped surrogate objective, which is widely-used in RL. We import the solution to our context, leading to the following practical procedure of generator updates.

**Probability Ratio Clipping.** Let $r_t$ denote the probability ratio $r_t(\boldsymbol{\theta}) = \frac{p_\theta(\boldsymbol{x})}{p_{\theta^{(t)}}(\boldsymbol{x})}$ which measures the difference between the new and old generator distributions. For instance, $r_t(\boldsymbol{\theta}^{(t)}) = 1$. The clipped surrogate objective for updating the generator, as adapted from PPO, is:

$$\mathcal{L}^{CLIP}(\boldsymbol{\theta}) = \mathbb{E}_{p_\theta}\left[\min\left(r_t(\boldsymbol{\theta})f_\phi(\boldsymbol{x}),\ r_t^{clip}(\boldsymbol{\theta})f_\phi(\boldsymbol{x})\right)\right], \tag{5}$$

where $r_t^{clip}(\boldsymbol{\theta}) = \text{clip}\left(r_t(\boldsymbol{\theta}), 1-\epsilon, 1+\epsilon\right)$ clips the probability ratio, so that moving $r_t(\boldsymbol{\theta})$ outside of the interval $[1-\epsilon, 1+\epsilon]$ is discouraged. Taking the minimum puts a ceiling on the increase of the objective. Thus the generator does not benefit by going far away from the old generator.

Finally, to estimate the probability ratio $r_t(\boldsymbol{\theta})$ when $p_\theta$ is implicit, we use an efficient approximation similar to [7, 14] by introducing a binary classifier $C$ trained to distinguish real and generated samples. Assuming an optimal classifier $C$ which has $p_\theta(\boldsymbol{x}) = \frac{1-C(\boldsymbol{x})}{C(\boldsymbol{x})}p_r(\boldsymbol{x})$ [7, 13], we approximate $r_t$ by:

$$r_t(\boldsymbol{\theta}) = \frac{p_\theta(\boldsymbol{x})}{p_{\theta^{(t)}}(\boldsymbol{x})} \approx \frac{(1-C(\boldsymbol{x})) \cdot C^{(t)}(\boldsymbol{x})}{(1-C^{(t)}(\boldsymbol{x})) \cdot C(\boldsymbol{x})}, \tag{6}$$

where $C^{(t)}(\boldsymbol{x})$ denotes the classifier at the $t$-th iteration. Note that the rightmost expression depends on $\boldsymbol{\theta}$ because $\boldsymbol{x}$ is the output of the generator, i.e., $\boldsymbol{x} = G_\theta(\boldsymbol{z})$. In practice, during the phase of generator training, we maintain $C$ by fine-tuning it for only one iteration every time after $\boldsymbol{\theta}$ is updated (Alg.1). Thus the maintenance of $C$ is cheap. We give more details of the configuration of $C$ in the appendix. In the cases where an explicit generative model is used (e.g., a language model for text generation), the probability ratio $r_t$ can directly be evaluated by definition without the need of $C$, though in our text generation experiments (Sec.4.2) we still used $C$ for approximating $r_t$.

## 3.3 Discriminator Training with Sample Re-weighting

We next discuss the training of the discriminator $f_\phi$, where we augment the conventional training with an importance weighting mechanism for adaptive updates. Concretely, given the form of the auxiliary distribution solution $q^{(t)}$ in Eq.(3), we first draw from the recent energy-based modeling work [24, 26] and propose to optimize $\phi$ by maximizing the data log-likelihood of $q^{(t)}$, $\mathcal{L}(\phi) = \mathbb{E}_{p_r}[\log q^{(t)}(\boldsymbol{x})]$. By taking the gradient, we have:

$$\nabla_\phi \mathcal{L}(\phi) = \nabla_\phi\left(\mathbb{E}_{p_r}\left[f_\phi(\boldsymbol{x})\right] - \log Z_\phi\right) = \mathbb{E}_{p_r}\left[\nabla_\phi f_\phi(\boldsymbol{x})\right] - \mathbb{E}_{q^{(t)}}\left[\nabla_\phi f_\phi(\boldsymbol{x})\right]. \tag{7}$$

We can observe that the resulting form resembles the conventional one (Eq.1) as we are essentially maximizing $f_\phi$ on real data while minimizing $f_\phi$ on fake samples. An important difference is that

here fake samples are drawn from the auxiliary distribution $q^{(t)}$ instead of the generator $p_\theta$. This difference leads to the new sample re-weighting component as below. Note that, as in WGAN (Sec.2), we maintain $f_\phi$ to be from the class of 1-Lipschitz functions, which is necessary for the convergence analysis in Sec.3.4. In practice, we can use gradient penalty [15, 52] for the Lipschitz continuity.

**Sample Re-weighting.** We use the tool of importance sampling to estimate the expectation under $q^{(t)}$ in Eq.(7). Given the multiplicative form of $q^{(t)}$ in Eq.(3), similar to [1, 11, 24], we use the generator $p_{\theta^{(t)}}$ as the proposal distribution. This leads to

$$\mathbb{E}_{q^{(t)}}\left[\nabla_\phi f_\phi(\boldsymbol{x})\right] = \mathbb{E}_{p_{\theta^{(t)}}}\left[\exp\{f_\phi(\boldsymbol{x})\} \cdot \nabla_\phi f_\phi(\boldsymbol{x})\right] / Z_\phi. \tag{8}$$

Note that $Z_\phi$ is the normalization factor defined in Eq.(3). Thus, fake samples from the generator are weighted by the exponentiated discriminator score when used to update the discriminator. Intuitively, the mechanism assigns higher weights to samples that can fool the discriminator better, while low-quality samples are downplayed to avoid destructing the discriminator performance. It is worth mentioning that similar importance weighting scheme has been used in [7, 23] for generator training in GANs, and [5] for improving variational auto-encoders. Our work instead results in a re-weighting scheme in the new context of discriminator training.

The algorithm below summarizes the proposed training procedure for the generator and discriminator.

---
**Algorithm 1** GAN Training with Probability Ratio Clipping and Sampling Re-weighting
---
1: Initialize the generator $p_\theta$, the discriminator $f_\phi$, and the auxiliary binary classifier $C$
2: **for** $t \leftarrow 1$ to $T$ **do**
3:     **for** certain number of steps **do**
4:         Update the discriminator $f_\phi$ with sample re-weighting through Eqs.(7)-(8), and maintain $f_\phi$ to have upper-bounded Lipschitz constant through, e.g., gradient penalty [15].
5:     **end for**
6:     **for** certain number of steps **do**
7:         Finetune the real/fake binary classifier $C$ (for 1 step)
8:         Estimate probability ratio $r_t(\boldsymbol{\theta})$ using $C$ through Eq.(6)
9:         Update the generator $p_\theta$ with probability ratio clipping through Eq.(5)
10:     **end for**
11: **end for**
---

## 3.4 Theoretical Analysis

To provide theoretical insight on the performance of our method, we prove that our framework holds the same guarantees as WGAN-GP [15] and Lipschitz GANs [59]. Formally, we show that the method is fully compatible with *Proposition 1* in [15] and *Theorem 2* in [59], which provides rigorous analysis on GANs with Lipschitz discriminators and concludes 1) informative gradient pushes the generator distribution to the real data distribution and 2) the only Nash-equilibrium is $p_\theta = p_r$. Note that the theorems do not guarantee distributional convergence of $p_\theta$ to $p_r$, same as in [15, 59].

Our analysis is based on the reverse KL updates for the generator (Eq.4), while the probability ratio clipping serves as a practical emulation for the updates. We begin by adapting *Proposition 1* in Gulrajani et al. [15] to our problem:

**Proposition 3.1.** *Let $p_r$ and $q$ be two distributions in $X$, a compact metric space. Then, there is a 1-Lipschitz function $f^*$ which is the optimal solution of*

$$\max_{\|f\|_L \le 1} \mathbb{E}_{\boldsymbol{x} \sim p_r}\left[f(\boldsymbol{x})\right] - \mathbb{E}_{\boldsymbol{x} \sim q}\left[f(\boldsymbol{x})\right]$$

*Let $\pi^*$ be the optimal coupling between $p_r$ and $q$, defined as the minimizer of: $W(p_r, q) = \inf_{\pi \in \Pi(p_r, q)} \mathbb{E}_{(\boldsymbol{x},\boldsymbol{y}) \sim \pi}\left[\|\boldsymbol{x} - \boldsymbol{y}\|\right]$ where $\Pi(p_r, q)$ is the set of joint distributions $\pi(\boldsymbol{x}, \boldsymbol{y})$ whose marginals are $p_r$ and $q$, respectively. Then, if $f^*$ is differentiable, $\pi^*(\boldsymbol{x} = \boldsymbol{y}) = 0$, and $\boldsymbol{x}_\tau = \tau \boldsymbol{x} + (1 - \tau)\boldsymbol{y}$ with $0 \le \tau \le 1$, it holds that $\mathbb{P}_{(\boldsymbol{x},\boldsymbol{y}) \sim \pi^*}\left[\nabla f^*(\boldsymbol{x}_\tau) = \frac{\boldsymbol{y} - \boldsymbol{x}_\tau}{\|\boldsymbol{y} - \boldsymbol{x}_\tau\|}\right] = 1$.*

Proposition 3.1 indicates that in presence of an optimal discriminator $f^*$, given any sample $\boldsymbol{y}$ drawn from the variational distribution $q$, there exists a sample $\boldsymbol{x}$ drawn from real data distribution $p_r$

| Method | IS (↑) | FID (↓) |
|---|---|---|
| Real data | 11.24±.12 | 7.8 |
| WGAN-GP (2017) | 7.86±.08 | - |
| CT-GAN (2018) | 8.12±.12 | - |
| SN-GANs (2018) | 8.22±.05 | 21.7±.21 |
| WGAN-ALP (2020) | 8.34±.06 | 12.96±.35 |
| SRNGAN (2020) | 8.53 ±.04 | 19.83 |
| Ours (re-weighting only) | 8.45±.14 | 13.21±.60 |
| Ours (full) | **8.69±.13** | **10.70±.10** |

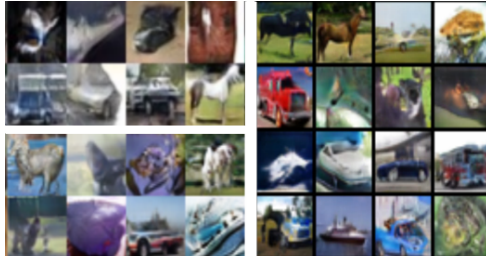

Table 1: CIFAR-10 results. Our method is run 3 times for average and standard deviation.

Figure 2: Generated samples by WGAN-GP (top-left), CT-GAN (bottom-left), and ours (right).

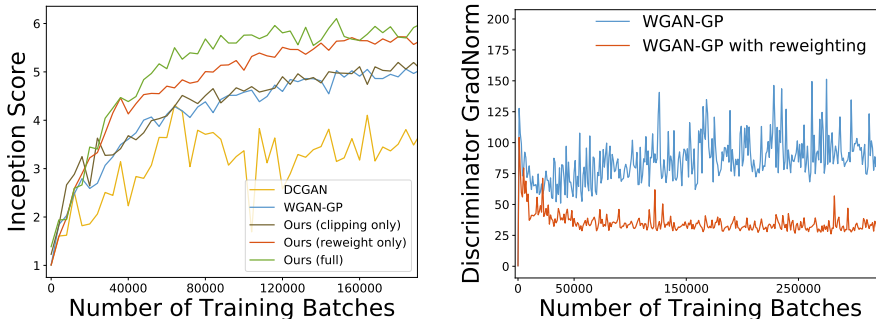

Figure 3: **Left:** Inception score on CIFAR-10 v.s. training batches (including both generator and discriminator batches). The DCGAN [39] architecture is used. **Right:** The gradient norms of discriminators on fake samples.

such that $\nabla_{\boldsymbol{x}} f^*(\boldsymbol{x}_\tau) = \frac{\boldsymbol{y} - \boldsymbol{x}_\tau}{\|\boldsymbol{y} - \boldsymbol{x}_t\|}$ for all linear interpolations $\boldsymbol{x}_\tau = \tau \boldsymbol{x} + (1 - \tau)\boldsymbol{y}$ with $0 \le \tau \le 1$. Therefore, an optimal discriminator $f^*$ can provide informative gradient to update $q$ and push $q$ towards to the real distribution $p_r$.

By the definition of $q$ with respect to $p_\theta$ in Eq.(3), the support of $p_\theta$ and $q$ are the same; namely, given any $\boldsymbol{x} \sim p_\theta, \boldsymbol{y} \sim p_r$, we also have $q(\boldsymbol{x}) \ne 0$. Therefore, for all $\boldsymbol{x} \sim p_\theta$, $\boldsymbol{x}$ is also a valid sample from $q$, the $f^*$ in Proposition 3.1 provides informative gradient with respect to $\boldsymbol{x}_\tau = \tau \boldsymbol{x} + (1 - \tau)\boldsymbol{y}, \forall \tau \in [0, 1]$: $\mathbb{P}_{(\boldsymbol{x}, \boldsymbol{y}) \sim \pi^*}\left[\nabla f^*(\boldsymbol{x}_\tau) = \frac{\boldsymbol{y} - \boldsymbol{x}_\tau}{\|\boldsymbol{y} - \boldsymbol{x}_\tau\|}\right] = 1$ Therefore, assuming $f^*$ is the optimal discriminator to (7), optimizing Eq.(4) can provide informative gradient that points the generator $p_\theta$ toward $p_r$.

## 4 Experiments

We conduct extensive experiments on three unsupervised generation tasks, including image generation, text generation, and text style transfer. The three tasks apply GANs to model different data modalities, namely, image, text, and neural hidden representations, respectively. Our approach consistently offers improvement over the state-of-the-arts on all tasks. See appendix for all experimental details.

### 4.1 Image Generation

We first use the popular CIFAR-10 benchmark for evaluation and in-depth analysis of our approach.

**Setup.** CIFAR-10 [28] contains 50K images of sizes $32 \times 32$. Following the setup in CT-GAN [52], we use a residual architecture to implement both generator and discriminator, and also impose a Lipschitz constraint on the discriminator. For each iteration, we update both generator and discriminator for 5 times. We use Inception Score (IS) [41] for evaluating generation quality and diversity, and Frechet Inception Distance (FID) [17] for capturing model issues, e.g., mode collapse [53].

**Results.** Table 1 reports the results on CIFAR-10. For the three latest methods, SN-GANs [33] introduced spectral normalization to stabilize the discriminator training; WGAN-ALP [49] developed an explicit Lipschitz penalty; and SRNGAN [42] introduced a weight-normalization scheme for generalization. Table 1 shows that our full approach (CT-GAN + discriminator sample re-weighting +

| Length | MLE | SeqGAN [56] | LeakGAN [16] | RelGAN [35] | WGAN-GP [15] | Ours | Real |
|--------|-----|-------------|--------------|-------------|--------------|------|------|
| 20 | 9.038 | 8.736 | 7.038 | 6.680 | 6.89 | **5.67** | 5.750 |
| 40 | 10.411 | 10.310 | 7.191 | 6.765 | 6.78 | **6.14** | 4.071 |

Table 2: Oracle negative log-likelihood scores ($\downarrow$) on synthetic data.

| Method | BLEU-2 ($\uparrow$) | BLEU-3 ($\uparrow$) | BLEU-4 ($\uparrow$) | BLEU-5 ($\uparrow$) | NLL$_{gen}$ ($\downarrow$) | Human ($\uparrow$) |
|--------|---------|---------|---------|---------|------------|---------|
| MLE | 0.768 | 0.473 | 0.240 | 0.126 | 2.382 | - |
| LeakGAN [16] | 0.826 | 0.645 | 0.437 | 0.272 | 2.356 | - |
| RelGAN 100 [35] | 0.881 | **0.705** | **0.501** | 0.319 | 2.482 | - |
| RelGAN 1000 [35] | 0.837 | 0.654 | 0.435 | 0.265 | 2.285 | 3.42±1.23 |
| WGAN-GP [15] | 0.872 | 0.636 | 0.379 | 0.220 | **2.209** | - |
| Ours | **0.905** | 0.692 | 0.470 | **0.322** | 2.265 | **3.59 ± 1.12** |

Table 3: Results on EMNLP2017 WMT News. BLEU measures text quality and NLL$_{gen}$ evaluates sample diversity. Results of previous text GAN models are from [35], where RelGAN (100) and RelGAN (1000) use different hyper-parameter for gumbel-softmax. Our approach uses the same gumbel-softmax hyper-parameter as RelGAN (1000).

generator probability ratio clipping) achieves the best, with both IS and FID significantly surpassing the baselines. These results accord with the visual results in Figure 2 where our generated samples show higher visual quality than those of the baselines. Comparison between CT-GAN and our approach with only re-weighting shows significant improvement. By further adding the probability ratio clipping to arrive our full approach, the performance (both IS and FID) is further improved with a large margin. The results demonstrate the effectiveness of the two components in our approach.

Figure 1 in Sec.1 has shown the effects of the proposed approach in stabilizing the generator and discriminator training. Here we further analyze these two components. Figure 3 (left) shows the convergence curves of different GAN methods. For a fair comparison, all models use the same DCGAN architecture [39], and both our approach and WGAN-GP [15] enforce the same discriminator Lipschitz constraint. Following the optimal setup in [15], the update ratio of both WGAN-GP and our "re-weighting only" is 5:1 (i.e., each iteration updates the discriminator for 5 times and the generator for one time). Our full approach and "clipping only" use an update ratio of 5:5, because the probability ratio clipping that discourages large generator updates allows us to update the generator more frequently, which is desirable. Note that the x-axis in Figure 3 accounts for both generator and discriminator batches (i.e., an 5:5 iteration is counted as 10 training batches). Thus, for any given point on the x-axis, all comparison methods used roughly the same amount of computation. From the curves, one can observe that our full approach surpasses our approach with only sample re-weighting, and they both converge faster and achieve a higher IS score than "clipping only", WGAN-GP, and DCGAN. It is interesting to note that "clipping only" does not offer a performance improvement over WGAN-GP, though its combination with sample re-weighting (i.e., the full approach) does improve over "re-weighting only". This is indeed not unexpected, because clipping and re-weighting are derived from the variational framework (Eq.2) in a principled way. Discarding either of the two could lead to improper handling of the variational distribution $q$ and fails to conform to the framework.

Figure 3 (right) investigates how the fake sample re-weighting can affect the discriminator training. By injecting re-weighting into WGAN-GP, the gradients on fake samples become more stable with lower variance, which partially explains the better training stability of discriminator in Figure 1.

## 4.2 Text Generation

In this section, we evaluate our approach on text generation, a task that is known to be notoriously difficult for GANs due to the discrete and sequential nature of text.

**Setup.** We implement our approach based on the RelGAN [35] architecture, a state-of-the-art GAN model for text generation. Specifically, we replace the generator and discriminator objectives in RelGAN with ours. We follow WGAN-GP [15] and impose discriminator Lipschitz constraint with gradient penalty. Same as [35], we use Gumbel-softmax approximation [25, 31] on the discrete text to enable gradient backpropagation, and the generator is initialized with maximum likelihood (MLE) pre-training. Same as previous studies, we evaluate on both synthetic and real text datasets.

| Method | BLEU |
|---|---|
| Zhang et al. [57] | 24.48 |
| Tian et al. [50] | 24.90 |
| Subramanian et al. [47] | 31.20 |
| Tikhonov et al. [51] | 32.82 |
| Ours | **33.45**±.95 |

Table 4: BLEU scores between model generations and human-written text on the Yelp data. We run our method for 5 times and report the average and standard deviation.

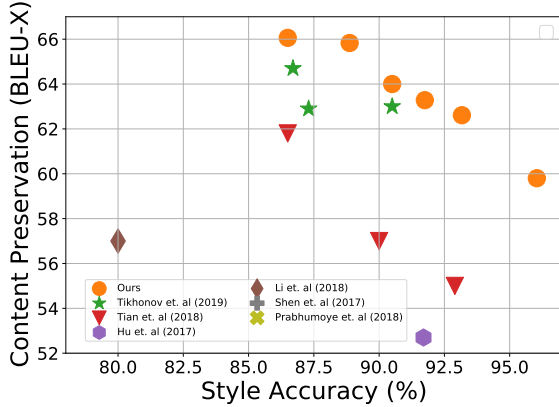

Figure 4: Trade-off between style accuracy and content preservation. The orange circles denote our results using varying values for an objective weight [51] which manages the trade-off.

**Results on Synthetic Data.** The synthetic data consists of 10K discrete sequences generated by an oracle-LSTM with fixed parameters [56]. This setup facilitates evaluation, as the quality of generated samples can be directly measured by the negative log-likelihood (NLL) of the oracle on the samples. We use synthetic data with sequence lengths 20 and 40, respectively. Table 2 reports the results. MLE is the baseline with maximum likelihood training, whose output model is used to initialize the generators of GANs. Besides the previous text generation GANs [16, 35, 56], we also compare with WGAN-GP which uses the same neural architecture as RelGAN and ours. From Table 2, one can observe that our approach significantly outperforms all other approaches on both synthetic sets. Our improvement over RelGAN and WGAN-GP demonstrates that our proposed generator and discriminator objectives are more effective than the previous ones.

**Results on Real Data.** We then evaluate our method on the EMNLP2017 WMT News, a large real text data used for text GAN studies [16, 35]. The dataset consists of 270K/10K training/test sentences with a maximum length of 51 and a vocabulary size of 5,255. To measure the generation quality, we use the popular BLEU-$n$ metric which measures $n$-gram overlap between generated and real text ($n \in \{2, 3, 4, 5\}$). To evaluate the diversity of generation, we use the negative log-likelihood of the generator on the real test set ($\mathrm{NLL}_{gen}$) [16, 35]. From the results in Table 3, one can see that our approach shows comparable performance with the previous best model RelGAN (100) in terms of text quality (BLEU), but has better sample diversity. Our model also achieves much higher BLEU scores than WGAN-GP. We perform *human* evaluation, with randomly sampled 50 sentences for RelGAN (1000) against ours and asked 5 annotators to score each sentence on a scale of 1-5. We use the same questions as designed by [35]. Ours obtained an average human score of $3.59 \pm 1.12$, higher than $3.42 \pm 1.23$ by RelGAN (Fleiss' Kappa score $0.61$ showing substantial inter-rater agreement).

### 4.3 Text Style Transfer

Text style transfer task is gaining increasing attention in NLP [22, 45, 54]. The task aims at rewriting a sentence to modify its style (e.g., sentiment) while preserving the content. Previous work applies GANs on neural hidden states to learn disentangled representations [45, 51]. The task thus can serve as a good benchmark for GANs, as hidden state modeling provides a new modality that differs from image and text modeling as studied above.

**Setup.** We follow the same experimental setting and use the same model architecture in the latest work [51]. In particular, the VAE-based model [22, 27] is extended by adding a latent code discriminator which eliminates stylistic information in the latent code. We replace their adversarial objectives with our proposed ones, and impose discriminator Lipschitz constraint with gradient penalty [15]. We test on sentiment transfer, in which the sentiment (positive/negative) is treated as the text style. We use the standard Yelp review dataset, and the ground truth output text provided by [30].

**Results.** Following the previous work [51], we first report the BLEU score that measures the similarity of the generated samples against the human written text. Table 4 shows that our approach achieves best performance, improving the state-of-the-art result [51] from BLEU 32.82 to 33.45.

The second widely used evaluation method is to measure (1) the style accuracy by applying a pre-trained style classifier on generated text, and (2) the content preservation by computing the BLEU score between the generated text and the original input text (BLEU-X). There is often a trade-off between the two metrics. Figure 4 displays the trade-off by different models. Our results locate on the top-right corner, indicating that our approach achieves the best overall style-content trade-off.

## 5 Conclusion

We have presented a new training framework of GANs derived from a new variational perspective and draws on rich connections with RL-as-inference. This results in probably ratio clipping for generator updates to discourage overly large changes, and fake sample re-weighting for stabilized discriminator updates. Experiments show our approach demonstrates superior training stability and improves over previous best methods on image generation, text generation, and text style transfer. The connection between the GAN and RL formalisms can potentially inspire more cross-pollination between the two fertile research fields. We are also interested in extending the formulation to connect more machine learning paradigms [21], for more systematic understanding, unification, and generalization of diverse learning algorithms.

## Broader Impacts

This work offers a unique viewpoint on two promising fields with lots of applications and impacts: Generative Adversarial Networks and Reinforcement Learning. The improvement to image generation results may be adapted to speed up photo editing, improve scene rendering, and create more realistic simulation for robot training. Furthermore, the contribution to text generation and text style transfer can be adopted to improve the quality of machine translation, and automated news-summaries.

Nevertheless, GANs can also be applied to faking images of people and jeopardize personal identities (i.e. Deepfake). We hope that future works can counter this issue through deep-fake detection.

## Footnotes

[1]Code available at: github.com/Holmeswww/PPOGAN

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
