[Supplementary Material · PPOGAN_NeurIPS_2020-2.pdf]

# 6 Appendix

## 6.1 Proof on the equivalence between Reverse KL Divergence and KL Divergence

We prove that optimizing $\mathbf{KL}(p_{\boldsymbol{\theta}}||q)$ are equivalent to optimizing $\mathbf{KL}(q||p_{\boldsymbol{\theta}})$. This provides guarantee for the approximation that leads to (4).

**Claim:** Under the assumption that $f_\phi$ Lipschitz, $f_\phi$ is bounded because the input $\boldsymbol{x}$ is bounded. Let $K$ be the Lipschitz constant of $f_\phi$, and let $c = f_\phi(0)$

$$|f_\phi(x) - c| \le K|x - 0| = K|x| \tag{9}$$

We then show that $\mathbf{KL}(p_{\boldsymbol{\theta}}||q)$ differ $\mathbf{KL}(q||p_{\boldsymbol{\theta}})$ by at most a constant. Since the function $f_\phi(\boldsymbol{x})$ is lower and upper-bounded. There exists $a, b$, such that $-a \le f_\phi(\boldsymbol{x}) \le b$ for any $\boldsymbol{x}$ bounded.

$$
\begin{aligned}
&\mathbf{KL}(q||p_{\boldsymbol{\theta}}) - \mathbf{KL}(p_{\boldsymbol{\theta}}||q) \\
&= \int_{\boldsymbol{x}} \left[ q(\boldsymbol{x}) \log\left(\frac{q(\boldsymbol{x})}{p_{\boldsymbol{\theta}}(\boldsymbol{x})}\right) - p_{\boldsymbol{\theta}}(\boldsymbol{x}) \log\left(\frac{p_{\boldsymbol{\theta}}(\boldsymbol{x})}{q(\boldsymbol{x})}\right) \right] d\boldsymbol{x} \\
&= \int_{\boldsymbol{x}} [q(\boldsymbol{x}) + p_{\boldsymbol{\theta}}(\boldsymbol{x})] \log\left(\frac{q(\boldsymbol{x})}{p_{\boldsymbol{\theta}}(\boldsymbol{x})}\right) d\boldsymbol{x} \\
&\overset{①}{=} \int_{\boldsymbol{x}} p_{\boldsymbol{\theta}}(\boldsymbol{x}) \left[1 + \frac{\exp\left(\alpha f_\phi(\boldsymbol{x})\right)}{Z}\right] \log\left(\frac{\exp\left(\alpha f_\phi(\boldsymbol{x})\right)}{Z}\right) d\boldsymbol{x} \\
&\overset{②}{\le} \alpha(a+b) \int_{\boldsymbol{x}} p_{\boldsymbol{\theta}}(\boldsymbol{x}) \left[1 + \frac{\exp\left(\alpha f_\phi(\boldsymbol{x})\right)}{Z}\right] d\boldsymbol{x} \\
&\overset{③}{=} 2\alpha(a+b),
\end{aligned}
\tag{10}
$$

where ① plugs $q^*(\boldsymbol{x}) = \frac{p_{\boldsymbol{\theta}}(\boldsymbol{x}) \exp\left(\alpha f_\phi(\boldsymbol{x})\right)}{Z}$; ② uses the fact $\log\left(\frac{\exp\left(\alpha f_\phi(\boldsymbol{x})\right)}{Z}\right) = \log\left(\frac{\exp\left(\alpha f_\phi(\boldsymbol{x})\right)}{\int_{\boldsymbol{x}} p_{\boldsymbol{\theta}}(\boldsymbol{x}) \exp\left(\alpha f_\phi(\boldsymbol{x})\right) d\boldsymbol{x}}\right) \le \log\left(\frac{\exp\left(\alpha b\right)}{\int_{\boldsymbol{x}} p_{\boldsymbol{\theta}}(\boldsymbol{x}) \exp\left(-\alpha a\right) d\boldsymbol{x}}\right) = \alpha(a+b)$; ③ uses $\int_{\boldsymbol{x}} p_{\boldsymbol{\theta}}(\boldsymbol{x}) \frac{\exp\left(\alpha f_\phi(\boldsymbol{x})\right)}{Z} d\boldsymbol{x} = 1$. The above claim completes the theoretical guarantee on the reverse-KL approximation in (**??**).

## 6.2 Proof on the necessity of Lipschitz constraint on the discriminator

Although [22] shows preliminary connections between PR and GAN, the proposed PR framework does not provide informative gradient to the generator when treated as a GAN loss. Following [58], we consider the training problem when the discriminator (i.e. $f_\phi(\boldsymbol{x})$ here) is optimal: when discriminator $f_\phi^*(\boldsymbol{x})$ is optimal, then the gradient of generator $g(f_\phi(\boldsymbol{x}))$ is $\nabla_{f_\phi^*(\boldsymbol{x})} g(f_\phi^*(\boldsymbol{x})) \cdot \nabla_{\boldsymbol{x}} f_\phi^*(\boldsymbol{x})$ which could be very small due to vanished $\nabla_{\boldsymbol{x}} f_\phi^*(\boldsymbol{x})$. In this way, it is hard to push the generated data distribution $p_{\boldsymbol{\theta}}$ towards the targeted real distribution $p_r$. This problem also exists in **??** because

$$f_\phi^*(\boldsymbol{x}) = \arg\min_{f_\phi(\boldsymbol{x})} \alpha\left[p_r(\boldsymbol{x})f_\phi(\boldsymbol{x}) - q(\boldsymbol{x})f_\phi(\boldsymbol{x})\right]. \tag{11}$$

So if $p_r$ and $q$ are disjoint, we have

$$
\begin{aligned}
f_\phi^*(\boldsymbol{x}) &= \arg\min_{f_\phi(\boldsymbol{x})} \alpha\left[p_r(\boldsymbol{x})f_\phi(\boldsymbol{x}) - q(\boldsymbol{x})f_\phi(\boldsymbol{x})\right] \\
&= \begin{cases} \arg\min_{f_\phi(\boldsymbol{x})} p_r(\boldsymbol{x})f_\phi(\boldsymbol{x}), & \text{if } \boldsymbol{x} \sim p_r \\ \arg\min_{f_\phi(\boldsymbol{x})} -q(\boldsymbol{x})f_\phi(\boldsymbol{x}), & \text{if } \boldsymbol{x} \sim q. \end{cases}
\end{aligned}
\tag{12}
$$

Note that for any $\boldsymbol{x} \sim p_r$, $f_\phi^*(\boldsymbol{x})$ is not related to $q$ and thus its gradient $\nabla f_\phi^*(\boldsymbol{x})$ also does not relate to $q$. Similarly, for any $\boldsymbol{x} \sim q$, $\nabla f_\phi^*(\boldsymbol{x})$ does not provide any information of $p_r$. Therefore, the proposed loss in [22] cannot guarantee informative gradient [58] that pushes $q$ or $p_{\boldsymbol{\theta}}$ towards to $p_r$.

## 6.3 Experiments: More Details and Results

### 6.3.1 Binary classifier for probability ratio clipping

For the image generation and text generation, the binary classifier $C$ in Eq.(6) has the same architecture as the discriminator except an additional Sigmoid activation at the output layer. The binary

classifier is trained with real and fake mini-batches alongside the generator, and requires no additional loops. We select the clipping parameter $\epsilon$ from $\{0.2, 0.4\}$, as they are typically used in PPO.

In addition in the task of image generation, we observe similar overall performance between training on raw inputs from the generator/dataset and training on input features from the first residual block of the discriminator ($D$), thus further reducing the computational overhead of the binary classifier.

### 6.3.2   Image Generation on CIFAR-10

We translate the code[2] provided by Wei et al. [50] into Pytorch to conduct our experiments. We use the same architecture: a residual architecture for both generator and discriminator, and enforcing Lipschitz constraint on the discriminator in the same way as CT-GAN [50]. During training, we interleave 5 generator iterations with 5 discriminator iterations. We optimize the generator and discriminators with Adam (Generator lr: $5e - 5$, Discriminator lr: $1e - 4$, betas: $(0.0, 0.9)$). We set the clipping threshold $\epsilon := 0.4$ for the surrogate loss and we linearly anneal the learning rate with respect to the number of training epochs.

**Discriminator sample re-weighting stabilizes DCGAN**   We quantitatively evaluate the effect of discriminator re-weighted sampling by comparing DCGAN [37] against DCGAN with discriminator re-weighting. Starting from the DCGAN architecture and hyper-parameters, we run 200 random configurations of learning rate, batch size, non-linearity (ReLU/LeakyReLU), and base filter count (32, 64). Results are summarized in Table 5. DCGANs trained with re-weighted sampling has significantly less collapse rate, and achieves better overall performance in terms of Inception Score. These results well demonstrate the effectiveness of the proposed discriminator re-weighted sampling mechanism.

| Method | Collapse rate | Avg IS | Best IS |
|---|---|---|---|
| DCGAN | 52.4% | 4.2 | 6.1 |
| DCGAN + Re-weighting | 30.2% | 5.1 | 6.7 |

Table 5: Outcomes of 200 trials with random configurations. The performance of the models are measured through Inception score. We identify training collapse when the average discriminator loss over 2000 batches is below $1e^{-20}$ or above $1 - 1e^{-20}$. DCGAN re-weighted with our loss has lower collapse rate and higher average performance.

**Discriminator re-weighted samples**   To provide an illustration of how discriminator weights can help the discriminator concentrate on the fake samples of better quality during the training phase, in Figure 5 we plot the fake samples of a trained ResNet model alongside their corresponding discriminator weights.

Figure 5: One batch of generated images together with their corresponding softmax discriminator weights. The more photo-realistic images (columns 2, 3, 5, 8) receive higher discriminator weights. In this batch, the generator will be influenced more by gradients from the better-quality samples above.

**Clipped surrogate objective**   One unique benefit of the clipped surrogate objective is that it allows our model to obtain an estimate of the effectiveness of the discriminator, which then enables us to follow a curriculum that takes more than one ($n_g$) generator steps per ($n_c$) critic steps. In practice, setting $n_g = n_c = 5$ achieves good quality, which also allows us to take 5 times more generator steps than prior works [2, 15, 31, 50] with the same number of discriminator iterations. Table 1 shows the improvement enabled by applying the surrogate objective.

**Generated samples**   Figure 6 shows more image samples by our model.

Figure 6: More samples from our generator on CIFAR-10

### 6.3.3 Text Generation

We build upon the Pytorch implementation[3] of RelGAN. We use the exact same model architecture as provided in the code, and enforce Lipschitz constraint on the discriminator in the same way as in WGAN-GP [2].

During training, we interleave 5 generator iterations with 5 discriminator iterations. We use Adam optimizer (generator lr: 1e-4, discriminator lr: 3e-4). We set the clipping threshold $\epsilon = 0.2$ for the surrogate loss and we linearly anneal the learning rate with respect to the number of training epochs.

### 6.3.4 Text Style Transfer

We build upon the Texar-TensorFlow [19] style-transfer model by Tikhonov et al. [49][4]. We use the exact same model architecture and hyper-parameters as provided in the code, and enforce Lipschitz constraint on the discriminator in the same way as WGAN-GP [2]. In addition, we replace the discriminator $D$ in Figure 7, by our loss with an auxiliary linear style classifier as in Odena et al. [36].

Figure 7: Model architecture from [49], where the style discriminator ($D$) is a structured constraint the generator optimize against. A latent code discriminator ensure the independence between semantic part of the latent representation and the style of the text. Blue dashed arrows denote additional independence constraints of latent representation and controlled attribute, see [49] for the details.

We did not apply the surrogate loss to approximate the KL divergence, but relied on gradient clipping on the generator.

## Footnotes

[2]github.com/biuyq/CT-GAN

[3]github.com/williamSYSU/TextGAN-PyTorch

[4]https://github.com/VAShibaev/text_style_transfer