[Reviews · NeurIPS 2020]

Review 1

Summary and Contributions: This work proposes to address the problem of training instability in Generative Adversarial Networks (GANs) by introducing two new techniques: probability ratio clipping (PRC) to stabilize generator training, and sample re-weighting to stabilize discriminator training. PRC is inspired by stabilization techniques used in reinforcement learning, and is used to ensure that updates to the generator do not change the weights too much. PRC is implemented by introducing a real/fake classifier (in addition to the standard discriminator) which is trained alongside the GAN and used to estimate probability ratios between the new and old distributions. Weight updates are clipped if the change in probability ratio is larger than some acceptable threshold. Sample re-weighting is a technique that has previously been applied in the GAN setting to adjust generator weight updates, but has not been applied to the discriminator. During the discriminator update, fake samples are weighted according to their discriminator scores such that realistic looking samples are up-weighted and low-quality samples are down-weighted. This technique reduces the potential for bad samples to produce large, inconsistent changes in the discriminator weights. Plots of training curves demonstrate that generator and discriminator losses are significantly more stable during training, and models trained with the proposed changes achieve better results than models which do not apply any additional stability scheme.

Strengths: S1 - Improving GAN training stability and generation quality is an important problem and of interest to many. S2 - Probability ratio clipping is well motivated by RL literature, and sample re-weighting from energy-based models. S3 - Proposed model outperforms all compared methods on multiple tasks: image generation, text generation, and text style transfer.

Weaknesses: W1 - Probability ratio clipping requires training an additional real/fake classifier, which increases training time and memory cost. W2 - Probability ratio clipping appears to achieve a very similar effect to using an exponential moving average (EMA) of weights, which is already a popular strategy for improving generator performance. It is unclear if probability ratio clipping has any benefit over EMA weights. See AF2 for more details.

Correctness: Co1 - Comparison to other methods seems fair. The proposed method is added to existing implementations with all other settings held constant, aside from the addition of a discriminator Lipschitz constraint via gradient penalty. It is unclear how much of the performance improvement can be attributed to the gradient penalty.

Clarity: Cl1 - Paper is well written and, for the most part, easy to understand. I appreciate the summaries after each equation block that describes their relevance.

Relation to Prior Work: RPW1 - There is some prior work that was released on Arxiv in February that is similar to the proposed sample re-weighting strategy [1]. They found that discarding the top-k worst generated samples in a batch when updating the discriminator improves performance, which agrees with the results shown here. Since this work is unpublished and was released close to the submission deadline I don't think it is necessary to compare to it, but it is relevant and could be mentioned as concurrent work. [1] Sinha, Samarth, et al. "Top-K Training of GANs: Improving Generators by Making Critics Less Critical." arXiv preprint arXiv:2002.06224 (2020).

Reproducibility: Yes

Additional Feedback: AF1 - The paper shows results of the model trained with re-weighing only, and then re-weighing + probability ratio clipping. I am curious what the performance of the model would be with only probability ratio clipping. Additionally, it would be good to include the performance of the baseline model trained without re-weighting or probability ratio clipping as a reference to indicate how much improvement can be attributed to the proposed additions. AF2 - The weight clipping strategy seems to be very similar in spirit to using an exponential moving average (EMA) of weights in the generator [2]. Weight EMA is commonly used in state-of-the-art models such as BigGAN [3] and StyleGAN [4]. Both weight clipping and weight EMA limit the amount of change that can be applied to the generator weights during an update, but they do so in different ways. Probability weight clipping performs a hard threshold on the weight update, while weight EMA performs an interpolation between the old and new weights. Another major difference is that weight EMA does not require the training of an additional network for probability ratio estimates. If the authors could comment on why one should spend the additional computation to use probability weight clipping over weight EMA, or even better, show experimental result comparing the two methods, I would be more convinced. (An implementation of weight EMA can be found in the BigGAN PyTorch repo: https://github.com/ajbrock/BigGAN-PyTorch/blob/master/utils.py#L614) AF3 - Broader impact statement does not discuss potential negative ethical and societal implications of the work. Overall, I think this is a good, well written paper. Discriminator sample re-weighing is a simple yet effective technique that appears to produce a significant boost in performance. Due to the ease of implementation I could see it being widely used. My main concern with this paper is the usefulness of the proposed probability weight clipping compared to existing methods. Currently, I suspect that it achieves that same effect as weight EMA while being more computationally expensive, but I hope the authors can show otherwise. [2] Yaz, Yasin, et al. "The Unusual Effectiveness of Averaging in GAN Training." International Conference on Learning Representations. 2018. [3] Brock, Andrew, Jeff Donahue, and Karen Simonyan. "Large scale gan training for high fidelity natural image synthesis." arXiv preprint arXiv:1809.11096 (2018). [4] Karras, Tero, Samuli Laine, and Timo Aila. "A style-based generator architecture for generative adversarial networks." Proceedings of the IEEE conference on computer vision and pattern recognition. 2019. == Post Rebuttal == After reading the rebuttal and other reviewer's comments I have decided to maintain my original overall score, but reduce my confidence in it. I still think that this work provides a useful bridge between techniques commonly used in RL and improving stability in GAN training. At the same time, I am not very familiar with RL literature or text generation/style transfer. Other reviewers seemed to have some concerns with these parts, so I defer to their judgement for these topics.


Review 2

Summary and Contributions: This paper propose to adopt the technique in the RL literature (e.g. TRPO and PPO) for RL-based TextGANs. Specifically, the paper propose to apply probability ratio clipping that regularizes generator training to prevent excessively large updates, and a sample re-weighting mecha8 nism that stabilizes discriminator training by downplaying bad-quality fake samples. The paper provides convergence analysis and empirically evaluates the proposed methods on benchmark datasets.

Strengths: The proposed methods are well motivated and have good theoretical foundations by connecting the RL literature with GAN training from a variational perspective. The theoretical deviation of the proposed method makes sense to me. The paper evaluates the proposed method on three different tasks and shows promising results.

Weaknesses: (1) The novelty of the proposed method is not very significant given the method is already established in the RL community. (2) The experimental setting (e.g. model architecture) needs to be described more in detail (now they seem to be included in the appendix). (3) For text generation experiments, it is important to include human evaluation results for the quality of generated sentences because corpus level BLEU score is not an accurate metric for sentence quality. It is also helpful to include some text examples generated by the proposed method and compared baselines in the Appendix. (4) The improvement on text generation task and style transfer task seems not very significant. (5) For text generation experiments, the author used the implementation of RelGAN, it is worth specify whether the paper uses the same hyperparameter as RelGAN(100) or RelGAN(1000). Also, is the WGAN-GP baseline based on the same architecture and hyperparameter?

Correctness: The methods and claims are correct.

Clarity: The paper is overall well written.

Relation to Prior Work: The relation with prior work is clearly discussed.

Reproducibility: Yes

Additional Feedback: Missing citation: Self-Adversarial Learning with Comparative Discrimination for Text Generation ICLR 2020, Zhou W, Ge T, Xu K, et al. == posted after author response == I have read the authors' response and other reviews. The authors' response has addressed some of my concerns. I tend to maintain my initial score.


Review 3

Summary and Contributions: The authors proposed a new variational GAN training framework with two components including probability ratio clipping and a sample re-weighting mechanism that enjoys superior training stability.

Strengths: 1. A new variational GAN training framework. 2. A probability ratio clipping is to regularize generator training to prevent excessively large updates. 3. A sample reweighting mechanism is to stabilize discriminator training by downplaying bad-quality fake samples. 4. A theoretical analysis on the convergence of their approach.

Weaknesses: 1. This paper is not well written in multiple places. There are many typos and inaccurate wordings throughout the paper. 2. The authors overclaim their theoretical analysis. My point is that the arguments in Section 3.4 cannot ensure the convergence of Algorithm 1. 3. It is unclear how to choose some key tuning parameters, such as $\epsilon$. 4. The key steps in algorithm 1 is not clear.

Correctness: The algorithm is correct. The empirical methodology looks ok.

Clarity: The paper is badly written. To name a few, this reviewer gave a list of them as follows. 1. In abstract, please check "we propose", and (1) and (2). You also use (1) and (2) for equation numbers. 2. “ varied architechtures for image and text generation". 3. We also use (1) and (2) in Introduction. 4. Page 4. what are C(x) and C^{(t)}(x)? 5. Page 4. Have you ever defined ${\mathcal L}(\phi)$? 6. page 5. In algorithm 1, you use C as the auxiliary classifier, whereas you use C(x) in (6).

Relation to Prior Work: Yes.

Reproducibility: Yes

Additional Feedback:


Review 4

Summary and Contributions: This paper proposes (1) *probability ratio clipping* and (2) *sample reweighting* to stabilize the training procedure of Generative Adversarial Networks (GAN). The probability ratio clipping method is adapted from Proximal Policy Optimization (PPO) to control the step-size of generator's off-policy update. The sample reweighting mechanism is designed to help the discriminator focus on better samples.

Strengths: By the experiments on diverse tasks, including image generation, text generation, and text style transfer, the boosted performance is shown to support the effectiveness of proposed approaches.

Weaknesses: -- The idea of probability ratio clipping is from PPO, so the contribution of this paper is incremental. -- There are some issues about derivation, baseline, and evaluation. Please refer to "Correctness."

Correctness: -- There are many ways to stabilize the training. Using PPO is reasonable in RL. Because in RL, sampling can be expensive, the model has to leverage the sampled experience as much as possible. On the other hand, in GAN, sampling is not as expensive as in RL, so it is possible to stabilize the training by having a large number of samples. For example, in Masson d'Autume et al., 2019 (https://arxiv.org/abs/1905.09922), large batch size (that is, many samples) is used to stabilize the training. However, this paper does not compare the proposed approach with the approach using a large batch size. Ans: The authors claim that the approach can integrate with large-batch training. -- The claim in Appendix 6.1, "We then show that KL(pθ||q) differs KL(q||pθ) by a constant", is not accurate. The difference is *bounded by a constant* not *a constant*. Thus, the derivation based on this claim is not precise. (answered) -- The authors do not evaluate diversity on text generation. In text generation, the good performance on BLEU & NLL may come from model collapsing. Without showing the diversity, the reader cannot tell whether the proposed model truly improves text generation. (answered) Minor issue: -- In 4.2, the authors say that "We then evaluate our method on the EMNLP2017 WMT News, the largest real text data used for text GAN studies". *WikiText-103* is used by de Masson d'Autume et al., 2019 (https://arxiv.org/abs/1905.09922), which contains 4 million sentences, larger than EMNLP2017 WMT News. -- In Fig.3 Left, what is the update ratio of WGAN-gp? I cannot find it in the paper, so I assume it is 5:1, the best setup in the original paper. If this is the case, since the proposed approach is 5:5, the comparison in Fig.3 Left seems unfair. When the two methods see the same number of training batches, they use different computations. (answered)

Clarity: Some equations need more description: -- $\mathcal{L}_\phi$ in eq. (7) is undefined. -- How to compute $Z_\phi$ in eq (8) (which is used to update the discriminator) is not clear. Several typos and grammatical mistakes: -- Line 3: we propose -> We propose -- Line 64: the stabilize generator -> the stablized generator? stabilizing generator? - Line 105: $Z_\phi = \int_x p_{\theta^{(t)}}(x) \exp(f_\phi(x))$ misses $dx$.

Relation to Prior Work: Yes

Reproducibility: Yes

Additional Feedback: -- In Fig. 1, the performance of "ratio clipping" plus "reweighting" and "reweighting only" are reported. How is the performance of "ratio clipping only"? -- In text generation, tokens are sampled from an explicit language model, does it imply that we don't need the classifier $C$? (related to the argument starts from line 133) -- The claim in 3.4: the direction of gradient is not sufficient for the convergence on $p_\theta = p_r$. Please provide more derivation.

[Author Response · NeurIPS 2020]

We thank all the reviewers for their insightful and encouraging comments. We're encouraged by the reviewers'
appreciation that 1) our method is well-motivated through the RL literature (R1, R2); 2) our empirical results on
multiple tasks are comprehensive and promising (R1, R2, R4); and 3) the paper is well-written (R1, R2).

We emphasize that the **main technical novelty** of the paper lies in connecting GAN training with both TRPO/PPO and
importance sampling through a new *principled variational GAN formulation* (Sec.3.1), which makes it possible to
re-purpose probability ratio clipping and re-weighting for GAN training. We also devise an approximation technique to
enable probability ratio estimation for implicit generative models. Our **empirical contributions** include the studies on a
broad range of tasks (image generation, text generation, text style transfer) and our consistently improved performance.

For *theoretical analysis*, we show that the method is fully compatible with *Theorem 2* in [56] (ICML'19) which provides
rigorous convergence analysis on GANs with Lipschitz discriminators and concludes 1) informative gradient pushes the
model distribution to the real data distribution and 2) the only Nash-equilibrium is $p_{model} = p_{data}$.

**Reviewer #1:** Thanks for your positive comments on 1) our good motivation through RL and EBM, 2) improved
performance on all 3 tasks, and 3) good paper writing. We'll add discussions and fix all issues in revision.

* *EMA:* EMA and our approach are *orthogonal*. EMA/MA averages generator parameters over time *outside* the training
loop (Yazıcı et al., ICLR2019) to reduce the stochasticity of mini-batch training, and thus is independent of how GAN
is trained. Moreover, EMA has to counter the generator's distributional shift issue by tuning hyper-parameters (window
size and average ratio). Our work can come complementary to EMA by discouraging distribution shifts with the new
surrogate loss, and can potentially make EMA easier to use. It's interesting to study the combination in the future.

* *Correctness:* In submission, we already compared with WGAN-GP under the same settings: image generation in
Table1 and text generation in Table 3. So the contribution of gradient penalty is already ruled out for comparison. Since
WGAN-GP alone on style-transfer has mode collapse issue, we did not discuss it.

* *Ratio-clipping only:* We emphasize that re-weighting and probability ratio clipping (KL regularization) are derived
from the variational framework (Eq.2) in a *principled* way, from introduction of the variational distribution $q$. Discarding
either of the two leads to improper handling of $q$ and fails to conform to the framework (and the theoretical properties).
We reported results of "reweighting-only for ablation study (despite its mathematical inappropriateness).

**Reviewer #2:** Thanks for appreciating that our method is well-motivated with good theoretical foundations, and shows
promising results on all three tasks. We'll add details in appendix, discuss related work, and fix all other issues.

* *Human evaluation:* Thanks for the suggestion. Following the same setting in the RelGAN paper, we conducted
human evaluation to compare RelGAN(1000) and our method. Ours obtained an average human score of 3.59, higher
than 3.42 by RelGAN (Fleiss' Kappa score 0.61 showing *substantial* inter-rater agreement).

* *Hyperparameters:* Our method and WGAN-GP baseline use the same hyperparameter setting as RelGAN(1000)

**Reviewer #3:** We selected $\epsilon$ from $\{0.2, 0.4\}$, as they are typically used in PPO. We'll fix all other issues in the revision.

**Reviewer #4:** We first clarify for several concerns:
• In text generation, $NLL_{gen}$ measures *diversity* (Line.235). Our model has better diversity than RelGAN (Table.3).
• In appendix 6.1, we meant it's *bounded* by a constant. The overall correctness is not affected. Also, please refer to
the clarification of the theoretical analysis above. We will revise the statements for clarity.
• In Fig.3 (left), the update ratio of WGAN-GP is 5:1 (the best setup), the reweighting-only method used 5:1, and
our full method used 5:5. We clarify that both WGAN-GP and ours used the *same amount of computations* (i.e.,
a 5:1 iteration is counted as 6 training batches, and a 5:5 iteration as 10 batches). We will make this clearer. The
probability ratio clipping that discourages large generator updates allows us to update the generator more frequently.

* *PPO motivation and large-batch training:* Besides sample efficiency, PPO has a strong motivation/intuition to
discourage excessively large model updates [45]. This suits well for stabilizing the generator in GANs, as acknowledged
by R1 and R2. In practice, our surrogate loss achieves similar effect as the KL penalty in variational framework (Fig. 1
Left). The controlled update size also enables more frequent generator updates and better efficiency (Fig. 3 Left).

Large-batch training is effective for stabilization, but doesn't solve instability alone: Masson d'Autume et al. also used
techniques including dense rewards and discriminator regularization; BigGAN used spectral normalization, truncation,
and progressive scaling architecture. Our approach is orthogonal and can be combined with large-batch training.

* *Clarity:* As in Line.143 above Eq.(7), $\mathcal{L}_\phi$ is "the data log-likelihood of $q^{(t)}$ w.r.t $\phi$", where $q^{(t)}$ is defined in Eq.(3).
$Z_\phi$ in Eq.(8) is also estimated with importance sampling with $p_{\theta^{(t)}}$ as the proposal. We will make these clearer.

Please refer to our response to R1 for the clarification of "ratio clipping only".

In text generation we still used classifier $C$ despite the explicit model (though it's not necessary).

[Meta-Review · NeurIPS 2020]

This is an interesting case. The R3 with the negative comments did not pointed out any specific flaws of the paper, so his review can not be taken into account. The whole idea is found interesting and simple, and this formulation has not been tested before in the GAN literature. The numerical experiments are quite broad and promising. The paper is well written, so I would argue that it can be accepted.